# Prevalence of Sexualized Substance Use and Chemsex in the General Population and Among Women: A Systematic Review and Meta-Analysis of Cross-Sectional Studies

**DOI:** 10.3390/healthcare13080899

**Published:** 2025-04-14

**Authors:** Manshuk Ramazanova, Botagoz Turdaliyeva, Alfiya I. Igissenova, Maiya Zhakupova, Akmaral Sh. Izbassarova, Mariya Seifuldinova, Gulnaz Nurlybaeva, Raushan Yergeshbayeva, Indira Karibayeva

**Affiliations:** 1Department of Public Health and Social Sciences, Kazakhstan Medical University “KSPH”, Almaty 050060, Kazakhstan; m_ramazanova00@mail.ru; 2Kazakh Scientific Center of Dermatology and Infectious Diseases, Almaty 050002, Kazakhstan; bot.turd@gmail.com; 3Department of Microbiology and Virology, Asfendiyarov Kazakh National Medical University, Almaty 050012, Kazakhstan; igisenova.a@kaznmu.kz; 4Department of Public Health, Asfendiyarov Kazakh National Medical University, Almaty 050012, Kazakhstan; m.zhakupova@kaznmu.kz; 5Department of Physical Medicine and Rehabilitation, Sports Medicine, Asfendiyarov Kazakh National Medical University, Almaty 050012, Kazakhstan; aksha55@mail.ru (A.S.I.); zoya999999@mail.ru (G.N.); 6Department of General Surgery and Topographic Anatomy, Asfendiyarov Kazakh National Medical University, Almaty 050012, Kazakhstan; amanmaria75@gmail.com; 7Department of Nutrition, Asfendiyarov Kazakh National Medical University, Almaty 050012, Kazakhstan; raushanbogdanovna21@gmail.com; 8Department of Health Policy and Community Health, Jiann-Ping Hsu College of Public Health, Georgia Southern University, Statesboro, GA 30460, USA

**Keywords:** sexualized substance use, chemsex, prevalence, women, general population, public health, meta-analysis, substance use, sexual behavior, harm reduction

## Abstract

**Background and Objectives**: Sexualized substance use (SSU) and chemsex have garnered increasing attention in public health research, particularly among men who have sex with men (MSM). However, the prevalence and implications of these behaviors in the general population and among women remain underexplored. This systematic review and meta-analysis aimed to synthesize existing evidence on the prevalence of SSU and chemsex in the general population, with a specific focus on women. **Materials and Methods**: Following the Preferred Reporting Items for Systematic Reviews and Meta-Analyses guidelines, PubMed, ProQuest, Scopus, Web of Science, Cochrane, and PsycINFO were searched for studies published before 18 February 2025. The keywords included “chemsex”, “sexualized substance use”, “prevalence”, and “women”. Studies were included if they reported prevalence data on SSU or chemsex in the general population or among women. The exclusion criteria included studies focused exclusively on MSM, adolescents, or specific drug toxicity. The pooled prevalence estimates were presented using forest plots, and the heterogeneity was assessed using I^2^ statistics in RStudio (version 4.3.2). **Results**: The findings reveal that SSU and chemsex are significant phenomena, with pooled prevalences of 19.92% in the general population and 15.61% among women. The higher prevalence of SSU (29.40%) compared with chemsex (12.66%) in the general population suggests that substance use during sex is a broader behavioral pattern. Among women, the prevalence of chemsex was notably lower (3.50%) than SSU (25.78%). **Conclusions**: This study underscores that SSU and chemsex are significant public health concerns extending beyond the MSM community. The findings highlight the need for inclusive public health strategies that address these behaviors across the general population. Future research should focus on standardizing definitions, exploring contextual factors, and developing targeted interventions to mitigate associated risks, such as sexually transmitted infections, substance dependency, and mental health disorders.

## 1. Introduction

Engaging in new sexual experiences and seeking heightened sensations can increase the likelihood of individuals participating in risky sexual behaviors, which may pose health risks [1,2]. Sexualized substance use (SSU) and chemsex—the intentional use of psychoactive substances to enhance sexual experiences—have garnered increasing attention in public health research, particularly within the context of men who have sex with men (MSM) [3].

Chemsex, a combination of the words “chemical” and “sex”, originates from the London LGBTQ+ club scene, and its definition varies in the literature [4]. Typically, chemsex refers to the consumption of specific substances, often referred to as the “four chems”—mephedrone, gamma-hydroxybutyrate (GHB), gamma-butyrolactone (GBL), and methamphetamine—with the intention of prolonging, intensifying, or removing inhibitions during sexual activity with a particular mindset [3]. This practice typically occurs in the context of extended sexual encounters involving multiple partners, lasting several hours or even days [5]. Some authors also include ketamine and cocaine in their definition of chemsex [5]. Additionally, recent studies have expanded the use of the term to include individuals with various sexual orientations.

Although there is no consensus on the definition of chemsex [6,7], evidence suggests a significant difference between SSU and chemsex users [8]. SSU is the deliberate use of one or more psychoactive substances before or during sexual activity, with objectives that may include extending duration, diversifying practices, and enhancing the experience or performance [9]. Regardless of the definition, SSU and chemsex often involve polydrug use and are associated with altered decision-making processes, which can lead to unsafe sexual behaviors such as condomless sex and the sharing of syringes and other drug paraphernalia [10]. Consequently, individuals who engage in SSU and chemsex are more susceptible to sexually transmitted infections (STIs) including HIV, abuse, and non-consensual sexual activity [11,12].

For these reasons, numerous studies emphasize the need to consider this phenomenon a public health priority. The focus on chemsex among MSM has overshadowed the broader implications of SSU and chemsex in the general population, particularly among women. This oversight is concerning given the rising use of novel psychoactive substances (NPSs), polydrug use, and the potential for SSU to contribute to adverse health outcomes, including STIs, mental health disorders, and substance dependence [12,13]. Despite the growing prevalence of SSU and chemsex, there remains a significant gap in the literature regarding its epidemiology outside the MSM community, particularly among women. This gap highlights the importance of understanding the prevalence of SSU and chemsex beyond the MSM community.

The aim of this review is to systematically examine the current literature to determine the prevalence of SSU and chemsex in the general population and, specifically, among females. By identifying the extent of SSU and chemsex in diverse populations, we can better address the associated health risks and provide comprehensive support services. For the purposes of this study, we will adhere to the definitions of SSU and chemsex as used by the authors of the published studies.

## 2. Materials and Methods

### 2.1. Search Strategy, Study Selection, Data Collection, and Meta-Analysis

A systematic review of studies published before 18 February 2025, was conducted following the Preferred Reporting Items for Systematic Reviews and Meta-Analyses (PRISMA) guidelines [14]. The review protocol was submitted to PROSPERO (ID: CRD420250653893) after confirming that no similar reviews existed.

The following databases were searched: PubMed, ProQuest, Scopus, Web of Science, Cochrane, and PsycINFO via OVID. The filters applied included publication in English, publication in scholarly journals, and document types limited to articles, research articles, and early access articles. No restrictions were placed on the year of publication.

To define the search terms, a preliminary PubMed search was conducted to identify relevant keywords from the titles and abstracts of studies focusing on the prevalence of sexualized substance use and chemsex in the general population. Based on this preliminary search, the following keywords were used in the final search strategy: “chemsex” OR “chem sex” OR “sexualized substance use” OR “sexualized drug use” AND “prevalence” AND “women”. Further details on the search strategy are provided in Table 1.

Table 2 presents the eligibility criteria used to select articles in accordance with the Population, Intervention, Comparator, Outcome, and Study Design (PICOS) framework. The population included the general adult population and studies on women, while studies exclusively on men, sexual or gender minorities, female or male sex workers, and adolescents were excluded. The intervention criteria encompassed SSU, SDU, and chemsex, whereas studies focusing solely on alcohol use or the use of specific drugs, such as gamma-hydroxybutyrate (GHB) or methamphetamine, were excluded. No comparator was applicable. The outcome of interest was the number of individuals reporting SSU, including SDU and chemsex, out of the total assessed, while studies examining the toxicity, overdose, and abuse of specific drugs, such as GHB or methamphetamine, were excluded. This review included cross-sectional and observational studies, whereas reviews, abstracts, editorials, and commentaries, as well as studies published in languages other than English, were excluded.

The eligibility assessment and data collection were conducted in accordance with the PRISMA guidelines [14]. The PRISMA checklist for systematic reviews is available in Appendix A. Two independent researchers performed a standardized search (M.R. and I.K.). After searching all databases, the results were combined in Excel, and duplicates were removed. Only unique records were screened for relevance based on the titles and abstracts. In the final stage of the eligibility assessment, full-text articles were evaluated against the inclusion criteria, and the relevant data were extracted using a standardized data collection form. The extracted information included the first author’s last name, year of publication, country, study design, setting or assessment method, mean age, the definition of SSU, total population assessed, total population reporting SSU, total women assessed, total women reporting SSU, substances used, and sexual orientation (if provided). Two datasheets were compared and combined. Discrepancies in the study selection between the researchers were resolved through discussion with the third author (B.T.), and consensus was reached for all included studies.

The meta-analysis was conducted using the RStudio software (version 4.3.2) [15]. Two R packages, meta and metafor, were used to perform the meta-analysis of proportions. The pooled mean prevalence of SSU, along with its 95% confidence interval (CI), was estimated using a random-effects model [16]. The results of the model were visually displayed using forest plots. The heterogeneity was assessed by calculating I^2^ [17]. Potential sources of heterogeneity were explored through meta-regression analysis. To account for the potential influence of small sample size studies—particularly relevant in random-effects models where smaller studies receive relatively higher weights—we conducted sensitivity analyses including influence analysis and leave-one-out analysis. These steps helped identify whether any single study disproportionately affected the overall estimate. Additionally, publication bias was evaluated through funnel plot visualization and Egger’s test [16]. Subgroup analysis was conducted based on the type and definition of SSU reported: SSU use and chemsex.

### 2.2. Risk of Bias and Certainty of Evidence Evaluation

The risk of bias (quality) assessment was conducted using an adapted version of the Newcastle–Ottawa Scale (NOS) for cross-sectional studies [18]. The modified NOS evaluates each study based on six criteria, organized into three assessment groups: selection (three criteria), comparability (one criterion), and outcome (two criteria). Each criterion can earn up to one point, and the comparability criterion can receive a maximum of two points. This results in a total score range of 0 to 7, with higher scores indicating superior study quality. The risk of bias assessments were independently carried out by two authors after they had agreed on the assessment procedures. A third author calculated the inter-rater agreement between the two assessors. In this review, studies with scores of five points or more for cross-sectional studies were deemed to be of satisfactory quality and included in the systematic review [19,20].

Adhering to the guidelines from the *Cochrane Handbook for Systematic Reviews of Interventions*, we assessed the certainty of evidence using the Grading of Recommendations Assessment, Development, and Evaluation (GRADE) framework [21]. Furthermore, this assessment followed the procedures outlined in research notes on the evaluation of GRADE in systematic reviews [22]. The certainty of evidence was calculated in RStudio, using the “GRADE” package. This framework comprises five domains: risk of bias, assessed using the NOS for the cross-sectional studies checklist mentioned earlier; inconsistency, assessed via the I^2^ statistic; indirectness, assessed via PICO criteria; imprecision, assessed by determining if the 95% CI of the pooled estimate crosses the threshold of interest; and publication bias, assessed using Egger’s test results.

## 3. Results

### 3.1. Characteristics of Included Studies

A total of 1527 articles were identified using the search strategy described above. After removing duplicates, 1084 titles and abstracts were screened, of which 140 articles were selected for full-text evaluation. However, the full text was unavailable for one study. Following a full-text assessment, 13 studies met the PICOS eligibility criteria and were included in the meta-analysis. Among the excluded articles, 47 focused on a specific population, including 1 study that examined only adolescents [23], 44 that lacked the required data, and 18 that were not observational cross-sectional studies. Additionally, four studies examined the toxicity of a single substance [24,25,26,27], four presented data on drug-associated deaths [28,29,30,31], and three were healthcare professionals’ assessments [32,33,34]. The PRISMA flowchart illustrating the study selection and inclusion process is presented in Figure 1 [14].

Among the included studies, twelve presented data on the prevalence of SSU in the general population, while thirteen focused on the female population. Most of these studies were conducted in Europe. The assessment settings varied, with eight studies using online questionnaires, three studies involving STI clinic participants, and others including household surveys, as well as surveys among erotic show attendees. The definitions of SSU and chemsex also varied between studies. Overall, 38,359 individuals were questioned about SSU, with 17,112 respondents confirming its use. In the thirteen studies that assessed females only, 24,484 women were asked about SSU, and 7918 confirmed its use. Further details on the study participants can be found in Table 3.

### 3.2. Meta-Analysis of SSU Prevalence

The mean prevalence of SSU among the general population was 19.92% (95% CI: 13.81%; 27.86%). The prevalence estimate exhibited high heterogeneity: I^2^ = 100%, Q (df = 11) = 2948.02, and *p* < 0.001. The mean SSU prevalence estimates were higher among studies that focused on SSU (29.40% (95% CI: 21.17%; 39.24%)), also with high heterogeneity (I^2^ = 100%, Q (df = 5) = 1259.67, and *p* < 0.01). Conversely, the mean prevalence estimates were considerably lower in studies that focused on exclusively chemsex (12.66% (95% CI: 7.06%; 21.65%)), also with high heterogeneity (I^2^ = 98%, Q (df = 5) = 254.9, and *p* < 0.01), as presented in Figure 2a.

The mean prevalence of SSU among the female population was 15.61% (95% CI: 11.53%; 20.80%). The prevalence estimate exhibited high heterogeneity (I^2^ = 99%, Q (df = 12) = 1150.79, and *p* < 0.01). The mean SSU prevalence estimates were higher among studies that focused on SSU (25.78% (95% CI: 19.67%; 33.00%)), also with high heterogeneity (I^2^ = 99%, Q (df = 6) = 725.78, and *p* < 0.01). As is the case with the general population, the mean prevalence estimates were considerably lower in studies that focused on exclusively chemsex (3.50% (95% CI: 1.42%; 8.35%)), also with high heterogeneity (I^2^ = 91%, Q (df = 5) = 53.37, and *p* < 0.01), as presented in Figure 2b.

The heterogeneity of the pooled estimate among the general population was initially assessed using influence analysis and leave-one-out analysis. Both assessments identified the Miltz (2021) study [39], which had the lowest prevalence estimates, as the most influential study that significantly impacted the pooled mean prevalence of SSU in the general population (Figure 3a,b).

Furthermore, two meta-regression models were constructed to assess the heterogeneity of the pooled estimate among the general population. The first model evaluated the impact of the number of LGBTQ+ individuals on the pooled mean prevalence of SSU, while the second model examined the influence of the year of publication on the pooled mean SSU prevalence. Neither model revealed significant associations at the *p* < 0.05 cutoff point. The results are available in Appendix A.

Similarly, the heterogeneity of the pooled estimate among the female population was initially assessed using influence analysis and leave-one-out analysis. Both assessments identified the Miltz (2021) [39] and Gertzen (2024) [46] studies, which had the lowest prevalence estimates, as the most influential studies that significantly impacted the pooled mean prevalence of SSU in the female population (Figure 4a,b).

Furthermore, a meta-regression model was constructed to assess the heterogeneity of the pooled estimate among the female population, examining the influence of the year of publication on the pooled mean SSU prevalence. The meta-regression model did not reveal a significant association at the *p* < 0.05 cutoff point. The results are available in Appendix A.

Figure 5a,b display the funnel plots used to evaluate the publication bias for the pooled SSU prevalence estimates in the general population and the female population, respectively. Both plots show noticeable asymmetry. Furthermore, the presence of publication bias was confirmed by the significant results of Egger’s test (*p* < 0.05).

### 3.3. Risk of Bias (Quality) Assessment and Certainty of Evidence

All included cross-sectional studies had an NOS score of at least six out of seven. These scores indicate that the studies were of excellent quality and exhibited a low risk of bias, as demonstrated in Table 4.

The results from the GRADE certainty assessment, as presented in Table 5, suggest that the pooled mean SSU prevalence levels in both the general population and the female population had a low level of certainty. Therefore, these findings should be viewed and acted upon with caution.

## 4. Discussion

This systematic review and meta-analysis provide the first comprehensive synthesis of the prevalence of SSU and chemsex in the general population, with a particular focus on women. The findings reveal that SSU and chemsex are significant phenomena, with pooled prevalences of 19.92% in the general population and 15.61% among women. These estimates underscore the need to broaden the scope of research and intervention efforts beyond the MSM community, as SSU is evidently prevalent across diverse demographic groups. The higher prevalence of SSU (29.40%) compared with chemsex (12.66%) in the general population suggests that substance use during sex is a broader behavioral pattern, while chemsex represents a more specialized subset of this behavior. Among women, the prevalence of chemsex was notably lower (3.50%), with a wide range of 18% to 0%, highlighting the need for further research to understand the contextual and cultural factors that may influence women’s engagement in chemsex.

Our findings contribute to the growing body of literature on the prevalence of SSU and chemsex. A global analysis reported a mean prevalence of intoxicating substance use before or during sex at 37% (95% CI: 28%; 47%) among adults aged 18–29 years [48], highlighting significant engagement in this risk behavior. In comparison, our findings indicate a lower prevalence of SSU in the general population across studies, estimated at 20% (95% CI: 14%; 28%). Furthermore, a recent meta-analysis reported that 16% (95% CI: 11%; 21%) of MSM in Europe have engaged in chemsex [49], while another meta-analysis found that 19% (95% CI: 15%; 23%) of MSM in Asia have participated in chemsex [50]. Our research represents the first meta-analysis to aggregate data on SSU and chemsex in the general population. The observed prevalence of chemsex in our study (12.66% in the general population and 3.50% among women) is comparatively lower; however, these figures remain concerning and reinforce the notion that chemsex extends beyond the MSM community. These findings underscore the need for inclusive public health strategies that address SSU and chemsex across diverse populations.

An additional concerning trend in this study area is the rising prevalence of SSU and chemsex among adolescents. While we excluded one study reporting an SSU prevalence of 12% to 14% among adolescents in Spain [20], the available evidence suggests that this demographic is increasingly engaging in these behaviors. For example, a study on MSM adolescents in Brazil and Spain found a notably high prevalence of chemsex, with 31% (95% CI: 27%; 35%) reporting engagement in the practice [51]. Another study examining chemsex among adolescent and young MSM highlighted the significant influence of peer networks in fostering participation in chemsex [52]. These findings suggest that social dynamics and peer acceptance play a critical role in shaping adolescent substance use behaviors during sex. Given the well-documented risks associated with chemsex—including increased vulnerability to sexually transmitted infections, substance dependency, and mental health disorders—these patterns highlight the urgent need for targeted prevention and harm reduction strategies.

The literature reveals considerable variability in the definitions of SSU and chemsex, complicating direct comparisons across studies. SSU is broadly defined as the use of a wide range of illicit substances and alcohol in the context of sexual relationships, as presented in Table 1 of our research findings. Within SSU, chemsex is identified as a subculture involving the use of specific drugs to enhance sexual experiences, with one study even including 23 psychoactive substances in its definition of chemsex [40]. This lack of standardized definitions poses challenges in assessing the true prevalence and associated risks of these behaviors. Nevertheless, chemsex is associated with prolonged sexual interactions, where polydrug use can potentiate effects, ultimately leading to increased risks of sexually transmitted diseases and a broad range of mental health issues, including suicidal ideation and overdose [53,54,55]. Moreover, there is a documented risk of abuse and violation of mutual consent when the judgment of one partner—often the more vulnerable—is impaired by the psychoactive effects of stimulants, underscoring the ethical and safety concerns associated with chemsex and SSU [56,57,58].

Marked differences in the prevalence of SSU and chemsex across the United States and European countries underscore the powerful influence of sociocultural, legal, and temporal factors on these behaviors. Notably, the only US-based study included in this review was conducted in 2010 and reported no cases of SSU among women and a low overall prevalence [35]. This early timeframe likely reflects a different risk landscape, preceding the wider recognition of chemsex and the evolving patterns of SSU seen in more recent years. In contrast, more recent data from European countries—including Germany, the Netherlands, France, Spain, and Italy—demonstrate a notably higher prevalence of SSU and chemsex, including among women and heterosexual populations [59,60,61,62,63,64]. These differences may be shaped by more permissive or decriminalized drug laws, broader acceptance of recreational drug use within sexual subcultures, and stronger harm reduction frameworks that enable open disclosure. Furthermore, European STI clinics and online surveys may provide safer, more anonymous environments for participants to report sensitive behaviors compared with household surveys. These regional and temporal disparities highlight the importance of interpreting prevalence data within the sociopolitical and cultural context in which they were collected.

Several limitations warrant consideration. The heterogeneity among the included studies, particularly concerning the definitions of SSU and chemsex, assessment methods, demographic variations, and the primarily European origin of the included studies poses challenges to the generalizability of our findings. Additionally, the presence of publication bias may have influenced the pooled prevalence estimates. In our meta-regression analysis, we accounted for the number of non-heterosexual individuals in the studies; however, no significant association with SSU prevalence was observed. This finding suggests that while SSU and chemsex have been predominantly studied within MSM communities, these behaviors are not exclusive to any particular sexual orientation. Unfortunately, due to limited data, we were unable to conduct a similar analysis focusing solely on the female population. Future research should aim to explore the nuances of SSU and chemsex across different sexual orientations and gender identities, including adolescents. Populations beyond MSM remain understudied yet may exhibit distinct patterns of SSU and chemsex shaped by intersecting psychosocial, cultural, and developmental factors. Studies examining these subpopulations can provide deeper insight into risk trajectories and inform interventions tailored to their specific needs and vulnerabilities.

The findings of this review have several practical implications. Public health initiatives should broaden their focus to include diverse populations beyond MSM, recognizing that SSU and chemsex are behaviors present across various demographics. Educational programs on safe sexual behaviors should start early and target adolescents, given the high prevalence of SSU in this group and the associated risks of early sexual initiation and unprotected sex. Healthcare providers should be trained to identify and address SSU and chemsex behaviors in their patients, facilitating early intervention and support. Effective management of harmful chemsex should be multidisciplinary, addressing psychiatric, addictive, and infectious comorbidities, while emphasizing the importance of information and education to reduce associated risks and potential harm [65]. These interventions, including harm reduction programs, which have shown efficacy in addressing chemsex-related risks [66], should be expanded to reach underserved populations in this context, including women and adolescents. Sociodemographic, psychological, and behavioral factors likely shape both the initiation and continuation of these practices, including variations in motivations, risk perception, and engagement in protective behaviors. Future research is needed to explore these nuances and inform tailored interventions that consider the lived experiences and vulnerabilities of individuals engaged in chemsex and SSU.

## 5. Conclusions

This systematic review and meta-analysis revealed that SSU and chemsex are prevalent behaviors extending beyond the MSM community into the general population and among women, with a pooled prevalence of 19.92% in the general population and 15.61% among women. The high prevalence of SSU and chemsex among the general population and women underscores the need for comprehensive public health strategies and further research to understand the contextual factors influencing these behaviors. Addressing SSU and chemsex requires a multifaceted approach, incorporating education, prevention, and intervention efforts tailored to general populations to mitigate associated health risks. By broadening the scope of research and public health initiatives, we can better support individuals engaging in these behaviors and reduce the adverse outcomes linked to SSU and chemsex. However, given the methodological variability among included studies and the low certainty of evidence identified in our GRADE assessment, the findings of this meta-analysis should be interpreted and acted upon with caution.

## Figures and Tables

**Figure 1 healthcare-13-00899-f001:**
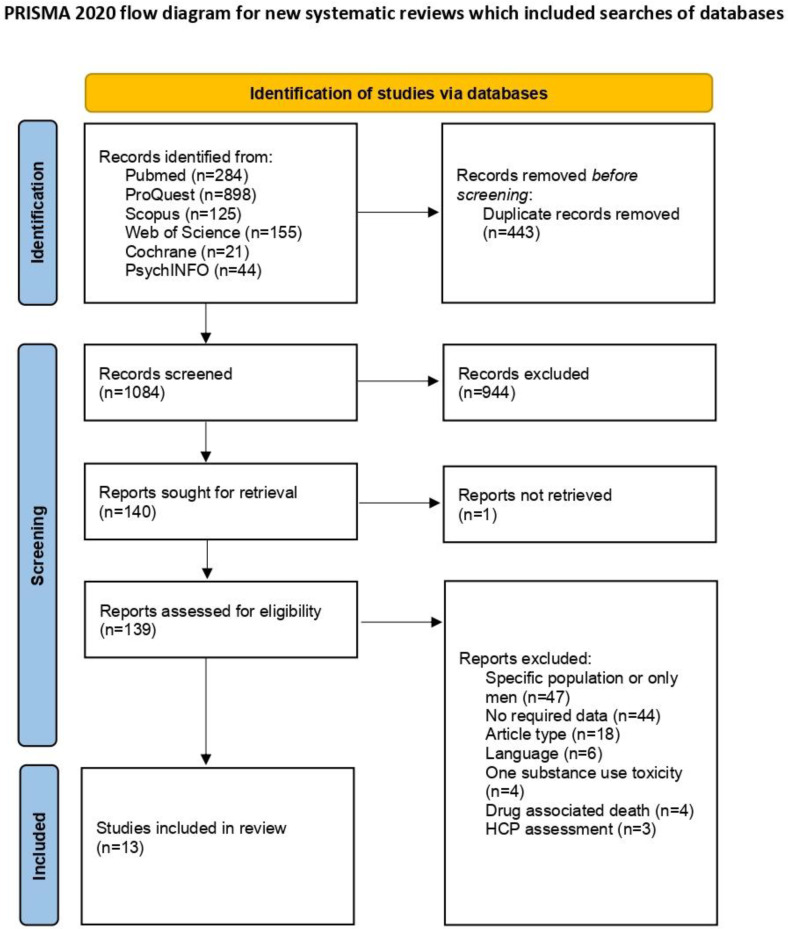
PRISMA flowchart of study inclusion.

**Figure 2 healthcare-13-00899-f002:**
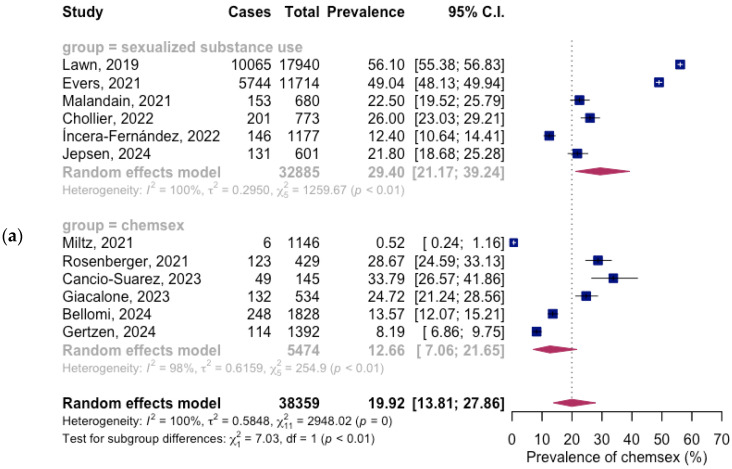
Forest plot of the pooled mean SSU prevalence: (**a**) general population; (**b**) female population [35,36,37,38,39,40,41,42,43,44,45,46,47].

**Figure 3 healthcare-13-00899-f003:**
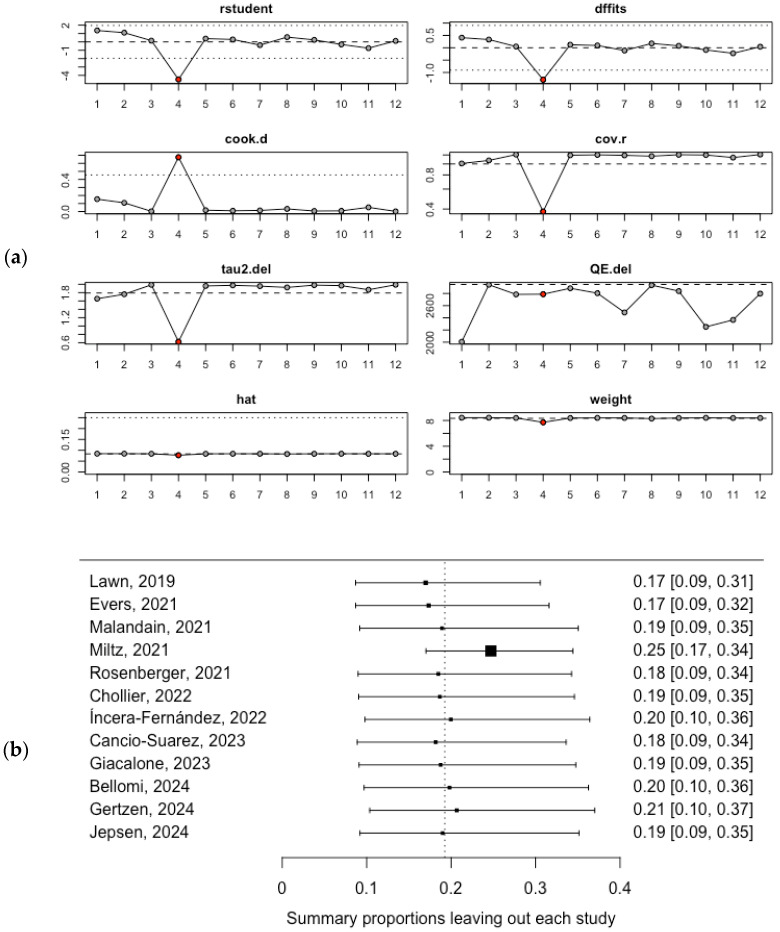
Heterogeneity assessment of the pooled mean SSU prevalence among the general population: (**a**) influence analysis; (**b**) leave-one-out analysis [36,37,38,39,40,41,42,43,44,45,46,47].

**Figure 4 healthcare-13-00899-f004:**
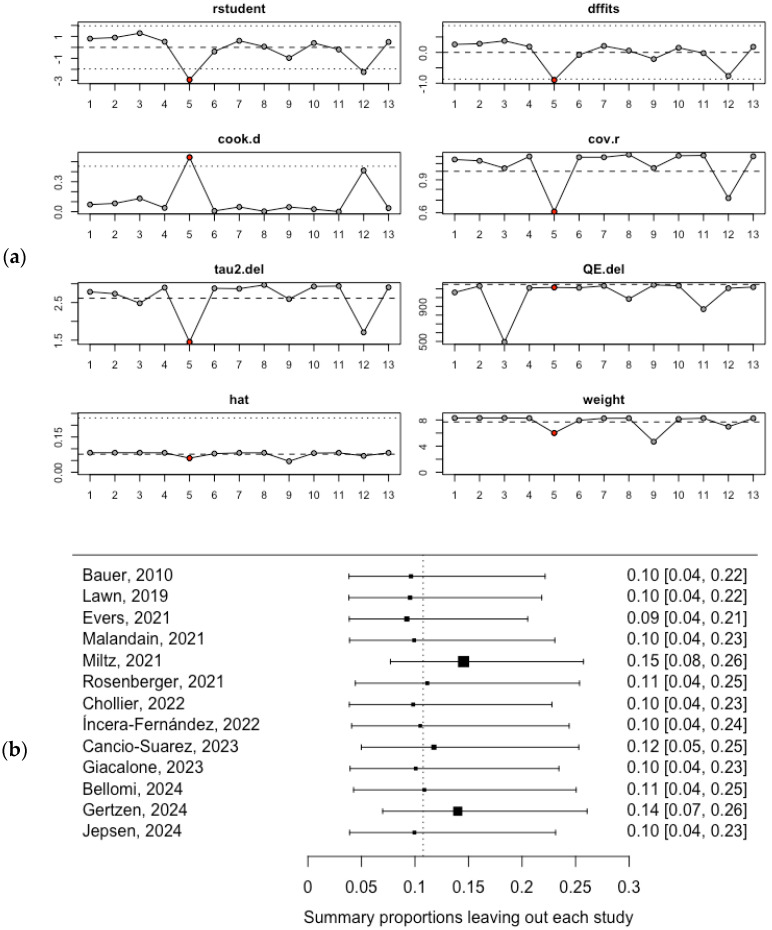
Heterogeneity assessment of the pooled mean SSU prevalence among the female population: (**a**) influence analysis; (**b**) leave-one-out analysis [35,36,37,38,39,40,41,42,43,44,45,46,47].

**Figure 5 healthcare-13-00899-f005:**
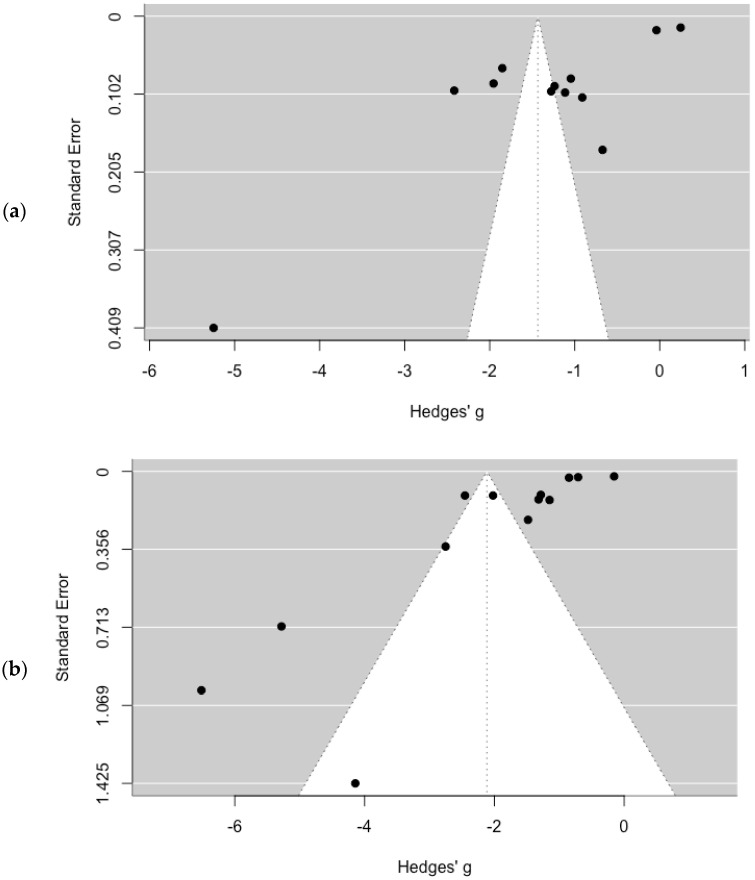
Funnel plots: (**a**) general population; (**b**) female population.

**Table 1 healthcare-13-00899-t001:** The search strategy used for the systematic review.

Database	Search Fields	Filters
PubMed	Title, Abstract, and Keywords	Language: English
ProQuest	Title, Abstract, and Keywords	Language: English Source type: Scholarly journals
Scopus	Title, Abstract, and Keywords	Language: English Document type: Articles
Web of Science	Title, Abstract, and Keywords	Language: English Document type: Articles and early access
Cochrane	Title, Abstract, and Keywords	No filter
PsychINFO via OVID	Sources: Journals, YourJournals, and Ovid Emcare	No filter

**Table 2 healthcare-13-00899-t002:** Inclusion and exclusion criteria of study selection based on the PICOS framework.

PICOS Framework	Inclusion Criteria	Exclusion Criteria
Population	General adult population; studies on women	Studies only on men, sexual or gender minorities, female or male sex workers, and adolescents
Intervention	Sexualized substance use, sexualized drug use, and chemsex	Alcohol or specific drug (GHB or methamphetamine) use only
Comparator	Not applicable	Not applicable
Outcome	Number of people reporting sexualized substance use, including sexualized substance use, sexualized drug use, and chemsex out of total assessed	Studies assessing the specific drug (GHB or methamphetamine) toxicity, overdose, and abuse
Study design	Cross-sectional, observational studies	Reviews, abstracts, editorials, and commentaries; also excluded studies published in languages other than English

**Table 3 healthcare-13-00899-t003:** Sexualized substance use study descriptions.

Last Name, Year	Design	Country	Group	Assessment Setting	SSU and Chemsex Definition	Age	Total/ Women	Cases/ Women	LGBTQ+ Total
Bauer, 2010 [35]	Cross-sectional	USA	SSU	Household survey	SSU: sex with male partner while under the influence of substances	28.7	0/5801	0/1737	625
Lawn, 2019 [36]	Cross-sectional	International	SSU	International online survey	SSU: most frequently used substances before or during sexual activity, including alcohol, cocaine, cannabis, GHB/GBL, ketamine, MDMA, mephedrone, methamphetamine, poppers, and sildenafil citrate (past 12 months)	31.4	17,940/6419	10,065/ 2117	3033
Evers, 2021 [37]	Cross-sectional	The Netherlands	SSU	Surveillance data analysis of STI clinic adults under 25	SSU: substance use, including alcohol and drugs, during sexual activity	Under 25	11,714/ 7761	5744/3580	N/A
Malandain, 2021 [38]	Cross-sectional	France	SSU	Online survey of university students	SSU: substance use during sexual activity		680/512	153/111	146
Miltz, 2021 [39]	Cross-sectional	England	Chemsex	Questionnaire of the heterosexual men and women	Chemsex: use of mephedrone, methamphetamine, or GHB/GBL (past 3 months)	Different age groups	1146/676	6/1	0
Rosenberger, 2021 [40]	Cross-sectional	Germany and German-speaking countries	Chemsex	Online survey in German-speaking countries	Chemsex: use of 23 psychotropic substances during sexual activity	35	429/150	123/9	257
Chollier, 2022 [41]	Cross-sectional	France	SSU	Paper survey of the erotic show attendees	SSU: use of alcohol, acids, cannabis, cocaine, crystal, ecstasy, and speed during sexual activity	34	773/320	201/77	106
Íncera-Fernández, 2022 [42]	Cross-sectional	Spain	SSU	Online survey of heterosexuals	SSU: drugs or substances before or during sexual activity	Various	1177/795	146/93	0
Cancio-Suarez, 2023 [43]	Cross-sectional	Spain	Chemsex	Online survey STI clinic patients	Chemsex: use of methamphetamine, GHB/GBL, mephedrone, cocaine, ketamine, poppers, speed, and other substances during sexual activity	36 (25–47)	145/31	49/0	87
Giacalone, 2023 [44]	Cross-sectional	Italy	Chemsex	Paper questionnaire at the STI clinic	Chemsex: use of methamphetamine, GHB/GBL, mephedrone, cocaine, ketamine, cannabis, poppers, sildenafil, and other substances during sexual activity	29	534/135	132/25	291
Bellomi, 2024 [45]	Cross-sectional	Italy	Chemsex	Online survey of country residents	Chemsex: use of specific psychoactive substances during sexual activity		1828/1124	248/89	186
Gertzen, 2024 [46]	Cross-sectional	Germany	Chemsex	Online survey	Chemsex: use of methamphetamine, GHB/GBL, mephedrone/cathinone, and ketamine during sexual activity		1392/395	114/2	899
Jepsen, 2024 [47]	Cross-sectional	Germany	SSU	Online survey of young adults	SSU: sex under the influence of illegal drugs	18–27	601/365	131/77	148

Abbreviations: GHB/GBL—gamma-hydroxybutyrate/gamma-butyrolactone; LGBTQ+—lesbian, gay, bisexual, transgender, queer/questioning, and others; MDMA—3,4-methylenedioxymethamphetamine; SSU—sexualized substance use; STI—sexually transmitted infections; USA—United States of America.

**Table 4 healthcare-13-00899-t004:** Risk of bias (quality) assessment according to the Newcastle–Ottawa Scale.

Study	Selection	Comparability	Exposure	Total
Bauer, 2010 [35]	4	1	2	7
Lawn, 2019 [36]	3	1	2	6
Evers, 2021 [37]	4	1	2	7
Malandain, 2021 [38]	2	1	2	6
Miltz, 2021 [39]	3	1	2	6
Rosenberger, 2021 [40]	4	1	2	7
Chollier, 2022 [41]	4	1	2	7
Íncera-Fernández, 2022 [42]	3	1	2	6
Cancio-Suarez, 2023 [43]	4	1	2	7
Giacalone, 2023 [44]	4	1	2	7
Bellomi, 2024 [45]	3	1	2	6
Gertzen, 2024 [46]	3	1	2	6
Jepsen, 2024 [47]	3	1	2	6

**Table 5 healthcare-13-00899-t005:** Evaluation of the certainty of evidence using GRADE framework.

Outcome	Study Design	Risk of Bias	Inconsistency	Indirectness	Imprecision	Publication Bias	Certainty of Evidence
Pooled mean SSU prevalence in the general population	Meta-analysis of cross-sectional studies	Low	Serious	Not serious	Not serious	Possible	Low
Pooled mean SSU prevalence in the female population	Meta-analysis of cross-sectional studies	Low	Serious	Not serious	Not serious	Possible	Low

## Data Availability

The original contributions presented in this study are included in this article. Further inquiries can be directed to the corresponding author.

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
