# Peer review of "Prevalence of Sexualized Substance Use and Chemsex in the General Population and Among Women: A Systematic Review and Meta-Analysis of Cross-Sectional Studies"

_healthcare, 2025, doi:10.3390/healthcare13080899_

Round 1
Reviewer 1 Report
Comments and Suggestions for Authors
The paper entitled “Prevalence of Sexualized Substance Use and Chemsex in the General Population and Among Women: A Systematic Review and Meta-Analysis of Cross-Sectional Studies” constitutes a remarkable effort to synthesize existing evidence on the prevalence of SSU and chem- sex in the general population, with a specific focus on women, a topic scare studied.
The manuscript is proper, it is clear and well-written, and the methods and results are straightforward. My only comment:
The authors mention that “evidence suggests a significant difference between SSU and chemsex users [lines 72-73].” However, the definition of SSU can be depicted more explicitly. For instance: Sexualized substance use (SSU) is the intentional use of one or more psychoactive substances before or during sexual activity, with objectives that may include extending duration, diversifying practices, and enhancing the experience or performance (Edmundson et al., 2018). I understand that is not a unique definition. However, I consider that making it more punctual can help people not related to the research topic.
- Edmundson, C., Heinsbroek, E., Glass, R., Hope, V., Mohammed, H., White, M., & Desai, M. (2018). Sexualised drug use in the United Kingdom (UK): A review of the literature. The International journal on drug policy, 55, 131–148. https://doi.org/10.1016/j.drugpo.2018.02.002
Author Response
We sincerely thank the reviewer for their invaluable feedback and constructive suggestions, which have significantly improved the quality and clarity of our manuscript. Our responses are in the attached document.

Reviewer 2 Report
Comments and Suggestions for Authors
Thank you very much to the authors for their text, it is interesting and valuable.
I would like to share with you some comments that are important to pay attention to:
1.My first doubt is regarding the number of authors. 9 authors for a systematic review is excessive. We need a good justification of their roles in the study.
2. Regarding the introduction, you need to clearly define what will be the perspective adopted on the concept of chemsex and SSU, as there is no consensus in the academic literature. They need to tell us their position in order to better understand their study.
3. You need to elaborate on the public health consequences, so that you can reinforce the relevance of the study.
4. Regarding the methodology, I consider that excluding adolescents, cis men and sexual minorities limits too much the possibility of generalizing the results. Please argue well the reason for this major limitation.
5. It is mentioned that the heterogeneity is high, but it is not justified in detail how this could affect the interpretation of the results.
6. I am very concerned about the sample size. It needs to be discussed and justified, as I consider that this conditions the statistical validity of the analysis.
7. Regarding the results, I am concerned that studies with varying definitions of SSU and chemsex were used, which could have biased the results of the meta-analysis.
8. I note that most of the studies are from Europe, so the findings may not be globally representative.
9. The discussion focuses on confirming that SSU and chemsex are common phenomena, but does not delve into contextual factors that might explain differences between studies.
10. There is no discussion of how these findings can inform public health intervention strategies.
11. In general, I consider that the study does not provide concrete recommendations based on its findings, and ends up being very descriptive. In addition, although the article emphasizes the relevance of the topic, there is no mention of how methodological limitations affect the extrapolation of the results.
Author Response

(The authors gave the same response as above.)

Reviewer 3 Report
Comments and Suggestions for Authors
This is an extraordinary and very timely article which begins to fill a gap in current knowledge concerning sexualized substance use and chemsex in two understudied populations, women and the general population. A very broad and contemporary literature review was effectuated with appropriate and well documented meta-analytic methods, with significant efforts made to reduce risk of bias and enhance the certainty of evidence.
The authors note that the majority of studies were done in Europe but that there is ample evidence suggesting that these are global phenomenon. Of more than 1084 records screened, ultimately only 13 were included in the final review. the authors point out the dangers associated with sexualized substance use and chemsex and note that the prevalence of chemsex in the general population was 19.92% and 15.61% in women and the prevalence of chemsex was 12.66% in the general population and 3.50% in women in the populations for which data was obtained. the authors also point out the problems with the definitions of these two entities. However, this study clearly presents information that suggests that these behaviors constitute a much broader public health phenomena than has previously been realized.
The authors are to be commended for this pioneering study.
Author Response
We sincerely thank the reviewer for their thoughtful and encouraging feedback on our manuscript.
Reviewer 4 Report
Comments and Suggestions for Authors
I am grateful for the opportunity to review the manuscript entitled: “Prevalence of Sexualized Substance Use and Chemsex in the General Population and Among Women: A Systematic Review and Meta-Analysis of Cross-Sectional Studies”.
After carefully reviewing your manuscript, I have identified several areas that require major revisions before it is suitable for publication.
Key Issues to Address:
- Systematic Review Methodology:
- PRISMA is mentioned, but a detailed compliance table should be included.
- It is unclear whether the Cochrane handbook was used for risk of bias assessment.
- The study selection process should be clarified and justified in more detail.
- Inclusion and Exclusion Criteria:
- Excluding studies on MSM limits the ability to compare with this population.
- The assessment of bias in studies reporting only SSU or chemsex is not clearly stated.
- Statistical Analysis:
- High heterogeneity (I² > 90%) raises concerns about the validity of pooled estimates.
- A more detailed discussion of factors contributing to heterogeneity is needed.
- The study mentions a random-effects model but does not address potential biases in small sample size studies.
- Risk of Bias and Evidence Quality:
- The Newcastle-Ottawa Scale is used, but it is unclear how studies with intermediate scores were classified.
- The GRADE assessment indicates low certainty, which should be emphasised further in the conclusions.
- Discussion and Conclusions:
- The discussion should be strengthened by addressing study limitations and potential biases.
- Comparing results with previous reviews would provide better context.
- More specific recommendations for future research should be outlined.
- References:
- Key references are included, but additional recent studies on the evolution of chemsex in different cultural contexts would enhance the review.
- Greater integration of literature on effective interventions for reducing SSU and chemsex-related risks is recommended.
Conclusion
This study makes a significant contribution to understanding the prevalence of SSU and chemsex in the general population and among women. However, methodological limitations, high heterogeneity, and low-certainty evidence weaken its impact. Future research should focus on standardising definitions, addressing selection bias, and implementing strategies to reduce heterogeneity in meta-analyses. The weaknesses mentioned above require changes to be made to the manuscript by the authors prior to publication.
Author Response

(The authors gave the same response as above.)

Round 2
Reviewer 2 Report
Comments and Suggestions for Authors
Many thanks to the authors for their efforts to enrich their article. I see all the suggested changes adopted. I would like to share with you a couple of additional mentions that I believe could benefit your work:
1. Although the justification you propose on methodology is sound, it could be briefly enriched with a note on how future research could address those excluded populations (adolescents, cis men, sexual minorities).
2. Regarding the contextual factors, the explanation you give does not fully address what was requested. It would be advisable to include at least one paragraph in the discussion on how sociocultural, legal or regional factors could explain the differences in prevalence (since in fact the document shows marked differences between countries). This would strengthen the interpretative dimension of the analysis.
Author Response
- Although the justification you propose on methodology is sound, it could be briefly enriched with a note on how future research could address those excluded populations (adolescents, cis men, sexual minorities).
Authors’ response: We thank the reviewer for their thoughtful comment. We have revised the section on future research to address this significant gap in our discussion. All changes are highlighted in lines 351-355. Future research should aim to explore the nuances of SSU and chemsex across different sexual orientations and gender identities, including adolescents. Populations beyond MSM remain understudied yet may exhibit distinct patterns of SSU and chemsex shaped by intersecting psychosocial, cultural, and developmental factors. Studies examining these subpopulations can provide deeper insight into risk trajectories and inform inter-ventions tailored to their specific needs and vulnerabilities.
- Regarding the contextual factors, the explanation you give does not fully address what was requested. It would be advisable to include at least one paragraph in the discussion on how sociocultural, legal or regional factors could explain the differences in prevalence (since in fact the document shows marked differences between countries). This would strengthen the interpretative dimension of the analysis.
Authors’ response: We thank the reviewer for their thoughtful comment. We have revised the discussion to address this point. All changes are highlighted in lines 324-339: Marked differences in SSU and chemsex prevalence across the United States and European countries underscore the powerful influence of sociocultural, legal, and temporal factors on these behaviors. Notably, the only U.S.-based study included in this review was conducted in 2010 and reported no cases of SSU among women and a low overall prevalence [35]. This early timeframe likely reflects a different risk landscape, preceding the wider recognition of chemsex and the evolving patterns of SSU seen in more recent years. In contrast, more recent data from European countries—including Germany, the Netherlands, France, Spain, and Italy—demonstrate notably higher SSU and chemsex prevalence, including among women and heterosexual populations [36–47]. These differences may be shaped by more permissive or decriminalized drug laws, broader acceptance of recreational drug use within sexual subcultures, and stronger harm reduction frameworks that enable open disclosure. Furthermore, European STI clinics and online surveys may provide safer, more anonymous environments for participants to report sensitive behaviors compared to household surveys. These regional and temporal disparities highlight the importance of interpreting prevalence data within the sociopolitical and cultural context in which it was collected.